Mutations of SARS-CoV-2 nsp14 exhibit strong association with increased genome-wide mutation load

http://orcid.org/0000-0003-0660-5791 Eskier Doğa 1 2
http://orcid.org/0000-0002-6872-9901 Suner Aslı 3
http://orcid.org/0000-0002-0158-2693 Oktay Yavuz 1 2 4 yavuz.oktay@ibg.edu.tr
http://orcid.org/0000-0001-6706-1375 Karakülah Gökhan 1 2 gokhan.karakulah@deu.edu.tr
1 Izmir Biomedicine and Genome Center , Izmir , Turkey
2 Izmir International Biomedicine and Genome Institute, Dokuz Eylül University , Izmir , Turkey
3 Department of Biostatistics and Medical Informatics, Faculty of Medicine, Ege University , Izmir , Turkey
4 Faculty of Medicine, Department of Medical Biology, Dokuz Eylül University , Izmir , Turkey
Bolshoy Alexander
Electronic publication date: 2020 Oct 12
Publication date: 2020
Volume: 8
Electronic Location ID: e10181
Received 2020 Aug 9; Accepted 2020 Sep 23
Copyright: © 2020 Eskier et al.
Copyright year: 2020
Copyright holder: Eskier et al.
License: This is an open access article distributed under the terms of the Creative Commons Attribution License, which permits unrestricted use, distribution, reproduction and adaptation in any medium and for any purpose provided that it is properly attributed. For attribution, the original author(s), title, publication source (PeerJ) and either DOI or URL of the article must be cited.
License URL: https://creativecommons.org/licenses/by/4.0/

Keywords: SARS-CoV-2, COVID-19, nsp14, 3′-5′ exonuclease, RNA-dependent RNA polymerase, RdRp, Mutation rate

Funding: Turkish Academy of Sciences Young Investigator Program (TÜBA-GEBİP) Yavuz Oktay is supported by the Turkish Academy of Sciences Young Investigator Program (TÜBA-GEBİP). The funders had no role in study design, data collection and analysis, decision to publish, or preparation of the manuscript.

==============================
SARS-CoV-2 is a betacoronavirus responsible for COVID-19, a pandemic with global impact that first emerged in late 2019. Since then, the viral genome has shown considerable variance as the disease spread across the world, in part due to the zoonotic origins of the virus and the human host adaptation process. As a virus with an RNA genome that codes for its own genomic replication proteins, mutations in these proteins can significantly impact the variance rate of the genome, affecting both the survival and infection rate of the virus, and attempts at combating the disease. In this study, we analyzed the mutation densities of viral isolates carrying frequently observed mutations for four proteins in the RNA synthesis complex over time in comparison to wildtype isolates. Our observations suggest mutations in nsp14, an error-correcting exonuclease protein, have the strongest association with increased mutation load without selective pressure and across the genome, compared to nsp7, nsp8 and nsp12, which form the core polymerase complex. We propose nsp14 as a priority research target for understanding genomic variance rate in SARS-CoV-2 isolates and nsp14 mutations as potential predictors for high mutability strains.

Introduction

COVID-19 is an ongoing global pandemic characterized by long-term respiratory system damage in patients, and is caused by the SARS-CoV-2 betacoronavirus. It is likely of zoonotic origin, but capable of human-to-human transmission, and since the first observed cases in the Wuhan province of China (Chan et al., 2020; Riou & Althaus, 2020), it has infected over 14 million people, with 612,054 recorded deaths (as of 22 July 2020). In addition to its immediate effects on the respiratory system, its long term effects are still being researched, including symptoms such as neuroinvasion (Li, Bai & Hashikawa, 2020; Wu et al., 2020), cardiovascular complications (Kochi et al., 2020; Zhu et al., 2020), and gastrointestinal and liver damage (Lee, Huo & Huang, 2020; Xu et al., 2020). Due to its high transmissibility, and capacity for asymptomatic transmission (Wong et al., 2020), study of COVID-19 and its underlying pathogen remain a high priority. As a result, the high amount of frequently updated data on viral genomes on databases such as GISAID (Elbe & Buckland-Merrett, 2017) and NextStrain (Hadfield et al., 2018) provides researchers with invaluable resources to track the evolution of the virus as it spreads across the world.

SARS-CoV-2 has a linear, single-stranded RNA genome, and does not depend on host proteins for genomic replication, instead using an RNA synthesis complex formed from nonstructural proteins (nsp) coded by its own genome. Four of the key proteins involved in the complex are nsp7, nsp8, nsp12, and nsp14, all of which are formed from cleavage of the polyprotein Orf1ab into mature peptides. Nsp12, also known as RdRp (RNA-dependent RNA polymerase), is responsible for synthesizing new strands of RNA using the viral genome as a template. Nsp7 and nsp8 act as essential co-factors for the polymerase unit, together creating the core polymerase complex (Kirchdoerfer & Ward, 2019; Peng et al., 2020), while nsp14 is an exonuclease which provides error-correcting capability to the RNA synthesis complex, therefore allowing the SARS-CoV-2 to maintain its large size genome (Subissi et al., 2014; Ma et al., 2015; Ogando et al., 2019; Romano et al., 2020). Owing to their role in maintaining replication fidelity and directly affecting the mutation-selection equilibrium of RNA viruses, these proteins are key targets of study in understanding the mutation accumulation and adaptive evolution of the virus (Eckerle et al., 2010; Peng et al., 2020).

In our previous study, we examined the top 10 most frequent mutations in the SARS-CoV-2 nsp12, and identified that four of them are associated with an increase in mutation density in two genes, the membrane glycoprotein (M) and the envelope glycoprotein (E) (the combination of which is hereafter referred to as MoE, as we previously described), which are under less selective pressure, and mutations in these genes are potential markers of reduced replication fidelity (Eskier et al., 2020a). In this study, we follow up on our previous findings and analyze the mutations in nsps 7, 8 and 14, in addition to nsp12, to identify whether the mutations are associated with a nonselective increase in mutation load or not. We then examine whole genome mutation densities in mutant isolates in comparison to wildtype isolates using linear regression models, in order to understand whether the mutations are associated with potential functional impact. Our findings indicate that mutations in nsp14 are most likely to be predictors of accelerated mutation load increase.

Materials and Methods

Genome sequence filtering, retrieval and preprocessing

As previously described (Eskier et al., 2020a), SARS-CoV-2 isolate genome sequences and the corresponding metadata were obtained from the GISAID EpiCoV database (date of accession: 17 June 2020). We applied further quality filters, including selecting only isolates obtained from human hosts (excluding environmental samples and animal hosts), those sequenced for the full length of the genome (sequence size of 29 kb or greater), and those with high coverage for the reference genome (<1% N content, <0.05% unique mutations, no unverified indel mutations). To ensure alignment accuracy, all nonstandard unverified nucleotide masking was changed to N due to the specifications of the alignment software, using the Linux sed command, and the isolates were aligned against the SARS-CoV-2 reference genome (NC_045512.2) using the MAFFT (v7.450) alignment software (Katoh et al., 2002), using the parameters outlined in the software manual for aligning closely related viral genomes (available at https://mafft.cbrc.jp/alignment/software/closelyrelatedviralgenomes.html). Variant sites in the isolates were annotated using snp-sites (2.5.1), bcftools (1.10.2), and ANNOVAR (release date 24 October 2019) software (Wang, Li & Hakonarson, 2010; Page et al., 2016), to identify whether a given mutation was synonymous or nonsynonymous. In addition, the 5′ untranslated region of the genome (bases 1–265) and the 100 nucleotides at the 3′ end were removed from the alignment and annotation files due to a high number of gaps and unidentified nucleotides. We further removed any sequences with incomplete sequencing location or date data in order to avoid complications in downstream analyses. Following the filters, 29,600 genomes were used for the analyses.

Mutation density calculation

Variants were categorized as synonymous and nonsynonymous following annotation by ANNOVAR, with intergenic or terminal mutations being considered synonymous. Gene mutation densities were calculated separately for synonymous and nonsynonymous mutations, as well as the total of SNVs, for each isolate, using a non-reference nucleotides per kilobase of region metric. Mutation densities were calculated for the combined membrane glycoprotein (M) and envelope glycoprotein (E) genes (MoE), the surface glycoprotein gene (S) and the whole genome.

Statistical analysis

Descriptive statistics for continuous variable days were calculated with mean, standard deviation, median, and interquartile range. Kolmogorov–Smirnov test was used to check the normality assumption of the continuous variables. In cases of non-normally distributed data, the Wilcoxon rank-sum (Mann–Whitney U) test was performed to determine whether the difference between the two MoE status groups was statistically significant. The Fisher’s exact test and the Pearson chi-square test were used for the analysis of categorical variables. The univariate logistic regression method was utilized to assess the mutations associated with MoE status in single variables, and then multiple logistic regression method was performed. The final multiple logistic regression model was executed with the backward stepwise method. The relationship between mutation density and time in isolates with mutations of interest, as well as in the group comprising all isolates, was examined via non-polynomial linear regression model and Spearman’s rank correlation. A p-value of less than 0.05 was considered statistically significant. All statistical analyses were performed using IBM SPSS version 25.0 (Chicago, IL, USA).

Results and Discussion

Increases in the mutation load of SARS-CoV-2 are unevenly distributed across its genome

To identify the trends in SARS-CoV-2 mutation load over time, we calculated the average mutation density per day for all isolates for whole genome, S gene and MoE regions, capping outliers at the 95th and 5th percentile values to minimize the potential effects of sequencing errors (Fig. 1). Our results show a very strong positive correlation between average mutation density and time, in both the genome level and the S gene. In comparison, MoE has a weak positive correlation, with a wider spread of mean density in early and late periods compared to the genome and the S gene. This is consistent with reduced selective pressure on the M and E genes, as has previously been described (Dilucca et al., 2020b). The top nonsynonymous mutation is 23403A>G (in 22,271 isolates), responsible for the D614G substitution in the spike protein, followed by the 14408C>T mutation (in 22,226 isolates) in the nsp12 region of the Orf1ab gene, causing P323L substitution in the RdRp protein, and the 28144C>T mutation (in 3,081 isolates), responsible for the L84S substitution in the Orf8 protein. The most common synonymous mutation is the 8782C>T mutation (in 3,047 isolates), and is found on the nsp4 coding region of the Orf1ab gene. For the S gene, the most frequent synonymous mutation is the 23731C>T mutation (in 622 isolates), and the second most common nonsynonymous mutation, after the aforementioned D614G mutation, is 25350C>T (in 215 isolates), responsible for the P1263L substitution. For MoE, the most common synonymous and nonsynonymous mutations are 26735C>T (in 341 isolates) and 27046C>T (in 530 isolates), respectively, both of which are found in the M gene, and the latter of which causes T175M amino acid substitution. Other than the D614G mutation, all of the mentioned mutations are C>T substitutions, the prevalence of which in T- or A-rich regions of the SARS-CoV-2 genome have been previously documented (Simmonds, 2020).

Figure 1 The average mutation density per day for genome, S gene and M and E genes.

(A) The mutation density vs. time for the whole SARS-CoV-2 genome. (B) The mutation density vs. time for the S gene. (C) The combined mutation density vs. time for the M and E genes. Values in y-axis represent the average number of SNVs in the corresponding day, normalized by kilobase of region of interest. SNV counts of genomes are normalized by capping at the 25-and 75-percentile values to minimize the effects of potential sequencing or assembly artifacts. Correlation scores are calculated using Spearman rank correlation.

Mutations in RNA synthesis complex proteins are associated with higher mutation load

After identifying the increase in mutation load over time, which was more prominent in genes with high functional impact (S, Orf1ab) compared to other structural genes (M, E and N), as seen in Fig. 1 and Figs. S1 and S2, we sought to examine possible associations of variants in proteins involved in SARS-CoV-2 genome replication with the increase. We first identified the five most frequently observed mutations for nsps 7, 8, 12 (also known as RdRp) and 14, four of the proteins cleaved from the Orf1ab polyprotein and are involved in the RNA polymerization, followed by analyzing the association of each mutation with the presence of MoE mutations (hereafter referred to as MoE status) using the chi-square test. A total of 12 out of the 20 mutations were found to have a significant association with MoE status (p-value < 0.05) (Table 1). Compared to our previous findings on the top 10 nsp12 mutations (Eskier et al. 2020a), which was based on an analysis of 11,208 samples as of 5 May 2020, 13536C>T and 13862C>T have increased in rank of appearance, from 6th and 7th to 4th and 5th, respectively, and decreased in p-value to show statistically significant associations. In addition, the 13730C>T mutation have increased in rank of appearance from 4th to 3rd. Out of the other nsps tested, nsp14 was found to have four significant mutations, while nsp7 had two and nsp8 had one.

Table 1 Comparisons of MoE and nsp mutations.

NSP	Mutations	Values	MoE absent	MoE present	Total	p	
n	%	n	%	n	%	
nsp7	11916C>T
S3884L	Absent	26,326	98.4	2,833	99.7	29,159	98.5	<0.001*	
Present	433	1.6	8	0.3	441	1.5	
12076C>T
N3937N	Absent	26,735	99.9	2,837	99.9	29,572	99.9	0.339	
Present	24	0.1	4	0.1	28	0.1	
11919C>T
S3885F	Absent	26,738	99.9	2,840	100.0	29,578	99.9	0.717	
Present	21	0.1	1	–	22	0.1	
12073C>T
D3936D	Absent	26,750	100.0	2,834	99.8	29,584	99.9	<0.001*	
Present	9	–	7	0.2	16	0.1	
11962C>T
L3899L	Absent	26,746	100.0	2,840	100.0	29,586	100.0	1.000	
Present	13	–	1	–	14	–	
nsp8	12478G>A
M4071I	Absent	26,757	100.0	2,750	96.8	29,507	99.7	<0.001*	
Present	2	–	91	3.2	93	0.3	
12550G>A
L4095L	Absent	26,697	99.8	2,841	100.0	29,538	99.8	–	
Present	62	0.2	–	–	62	0.2	
12415C>T
N4050N	Absent	26,725	99.9	2,841	100.0	29,566	99.9	–	
Present	34	0.1	–	–	34	0.1	
12557A>G
I4098V	Absent	26,729	99.9	2,841	100.0	29,566	99.9	–	
Present	30	0.1	–	–	30	0.1	
12400C>T
L4045L	Absent	26,734	99.9	2,840	100.0	29,574	99.9	0.508	
Present	25	0.1	1	–	26	0.1	
nsp12	14408C>T
P4715L	Absent	7,498	28.0	702	24.7	8,200	27.7	<0.001*	
Present	19,261	72.0	2,139	75.3	21,400	72.3	
14805C>T
Y4847Y	Absent	24,397	91.2	2,704	95.2	27,101	91.6	<0.001*	
Present	2,362	8.8	137	4.8	2,499	8.4	
13730C>T
A4489V	Absent	26,238	98.1	2,820	99.3	29,058	98.2	<0.001*	
Present	521	1.9	21	0.7	542	1.8	
13536C>T
Y4424Y	Absent	26,469	98.9	2,823	99.4	29,292	99.0	0.025*	
Present	290	1.1	18	0.6	308	1.0	
13862C>T
T4533I	Absent	26,535	99.2	2,833	99.7	29,368	99.2	0.001*	
Present	224	0.8	8	0.3	232	0.8	
nsp14	18060C>T
L5932L	Absent	25,247	94.3	2,768	97.4	28,015	94.6	0.001*	
Present	1512	5.7	73	2.6	1585	5.4	
18877C>T
L6205L	Absent	26,185	97.9	2,522	88.8	28,707	97.0	0.001*	
Present	574	2.1	319	11.2	893	3.0	
18998C>T
A6245V	Absent	26,454	98.9	2,836	99.8	29,290	99.0	0.001*	
Present	305	1.1	5	0.2	310	1.0	
18736T>C
F6158L	Absent	26,751	100.0	2,613	92.0	29,364	99.2	0.001*	
Present	8	–	228	8.0	236	0.8	
19524C>T
L6420L	Absent	26,530	99.1	2,825	99.4	29,355	99.2	0.102	
Present	229	0.9	16	0.6	245	0.8	
	Total	26,759	100.0	2,840	100.0	29,600	100.0		
Note:

* p-value < 0.05 was statistically significant.

Effects of geographical location on MoE status

In addition to time and genotype, we also examined the potential association between the location of isolates and MoE status as a possible confounding factor. We first examined whether there is a significant association between location, defined here as continent the isolate was originally obtained, and MoE status. Our results indicate that there is a strong association between location and MoE status, with the highest percentage of MoE present isolates in Asia (14.5%), and the percentage ratio in South America (6.5%) (p-value < 0.001). In comparison to our previous findings, South America had a dramatic decrease in MoE present isolate percentage, likely as a result of the increased sequencing efforts (from 118 isolates to 416) removing potential sampling biases or localized founder effects. Africa, Asia, and North America had an increase in MoE present proportion, while Europe, Oceania, and South America showed lowered percentages (Table 2).

Table 2 Distribution of MoE across geographical locations.

Locations	MoE absent	MoE present	Total	p	
	n	%	n	%	n	%		
Asia	2,319	85.5	394	14.5	2,713	100.0	<0.001*	
Africa	297	90.3	32	9.7	329	100.0	
South America	389	93.5	27	6.5	416	100.0	
Europe	14,879	89.8	1,697	10.2	16,576	100.0	
North America	7,401	93.4	522	6.6	7,923	100.0	
Oceania	1,474	89.7	169	10.3	1,643	100.0	
Total	26,759	90.4	2,841	9.6	29,600	100.0		
Note:

* p-value < 0.05 was statistically significant.

After observing the potential confounding effect of location on MoE status, we sought to understand whether a location is more or less likely to predict MoE status, using a logistic regression model (Table 3). Comparing each individual region (1) to the other five (0), we found that Asia, Europe and North and South America are all possible predictors of MoE status (p-value < 0.05), with Asia and Europe 1.697 and 1.184 times as likely to be MoE present as the other regions, and North and South America 0.589 and 0.650 times as likely, respectively.

Table 3 Logistic regression model of MoE and location on single variables.

Locations	p	OR	95% CI	
Asia	<0.001*	1.697	[1.513–1.903]	
Africa	0.937	1.015	[0.703–1.465]	
South America	0.032*	0.650	[0.439–0.963]	
Europe	<0.001*	1.184	[1.095–1.281]	
North America	<0.001*	0.589	[0.533–0.650]	
Oceania	0.330	1.085	[0.921–1.278]	
Notes:

* p-value < 0.05 was statistically significant.

OR, odds-ratio; CI, confidence interval.

Using these findings, we created different logistic regression models to identify which of the 12 mutations are likely to be independent predictors of MoE status (Table 4). In the single variable model, all 12 mutations we previously identified and location were found to be potential predictors (p-value < 0.05). Forming final models including the 12 mutations (Final Model A) and the mutations as well as locations (Final Model B), we observed that the predictor effect of two of the mutations nsp8 12478G>A and nsp14 18998C>T do not appear to be sufficiently independent of the other mutations in Final Model A. After adding the location variable to the Final Model A, location remains a significant predictor, with all five non-reference locations less likely to predict MoE than Asia, the reference location, and nsp12 14805C>T is found to not have a predictor effect independent of location (p-value = 0.073). Following Final Model B, nine mutations appear to have a significant association with MoE status, independent of other variables: 11916C>T, 12073C>T, 13536C>T, 13730C>T, 13862C>T, 14408C>T, 18060C>T, 18736T>C and 18877C>T (p-value < 0.05).

Table 4 Logistic regression model of MoE on single variables and a final model.

(Final Model A) Logistic regression model of ten mutations on final model. (Final Model B) Logistic regression model of four mutations and location on final model.

Mutations	Single variables	Final model A	Final model B	
	p	OR	95% CI	p	OR	95% CI	p	OR	95% CI	
Nsp7.11916	<0.001*	0.172	[0.085–0.346]	<0.001*	0.180	[0.089–0.363]	0.001*	0.314	[0.154–0.641]	
Nsp7.12076	0.403	1.571	[0.545–4.530]	–	–	–	–	–	–	
Nsp7.11919	0.433	0.448	[0.060–3.334]	–	–	–	–	–	–	
Nsp7.12073	<0.001*	7.341	[2.732–19.728]	<0.001*	8.108	[3.009–21.847]	<0.001*	9.164	[3.311–25.361]	
Nsp7.11962	0.756	0.724	[0.095–5.540]	–	–	–	–	–	–	
Nsp8.12478	<0.001*	442.707	[108.996–1,798.139]	–	–	–	–	–	–	
Nsp8.12550	0.997	–	–	–	–	–	–	–	–	
Nsp8.12415	0.998	–	–	–	–	–	–	–	–	
Nsp8.12557	0.998	–	–	–	–	–	–	–	–	
Nsp8.12400	0.388	0.377	[0.051–2.780]	–	–	–	–	–	–	
Nsp12.14408	<0.001*	1.186	[1.085–1.297]	<0.001*	1.310	[1.144–1.500]	<0.001*	1.662	[1.435–1.926]	
Nsp12.14805	<0.001*	0.523	[0.439–0.625]	0.007*	0.746	[0.603–0.923]	0.073	0.817	[0.655–1.019]	
Nsp12.13730	<0.001*	0.375	[0.242–0.581]	0.002*	0.497	[0.317–0.778]	<0.001*	0.393	[0.250–0.619]	
Nsp12.13536	0.026*	0.582	[0.361–0.938]	0.044*	0.611	[0.379–0.987]	0.009*	0.528	[0.327–0.855]	
Nsp12.13862	0.002*	0.335	[0.165–0.678]	0.004*	0.355	[0.175–0.720]	0.001*	0.293	[0.144–0.594]	
Nsp14.18060	<0.001*	0.440	[0.347–0.559]	0.001*	0.625	[0.479–0.816]	0.001*	1.658	[1.244–2.209]	
Nsp14.18877	<0.001*	5.770	[5.002–6.656]	<0.001*	5.543	[4.793–6.409]	<0.001*	6.437	[5.483–7.557]	
Nsp14.18998	<0.001*	0.153	[0.063–0.370]	–	–	–	–	–	–	
Nsp14.18736	<0.001*	291.773	[144.002–591.182]	<0.001*	368.884	[180.195–755.153]	<0.001*	970.884	[469.324–2,008.453]	
Nsp14.19524	0.104	0.656	[0.395–1.091]	–	–	–	–	–	–	
Location	<0.001*	–	–				<0.001*			
Africa	0.019*	0.634	[0.434–0.927]	–	–	–	0.017*	0.580	[0.391–0.860]	
South America	<0.001*	0.409	[0.273–0.612]	–	–	–	<0.001*	0.302	[0.198–0.461]	
Europe	<0.001*	0.671	[0.597–0.755]	–	–	–	<0.001*	0.681	[0.591–0.785]	
North America	<0.001*	0.415	[0.361–0.477]	–	–	–	<0.001*	0.228	[0.192–0.271]	
Oceania	<0.001*	0.675	[0.557–0.817]	–	–	–	<0.001*	0.536	[0.428–0.670]	
Notes:

* p-value < 0.05 was statistically significant.

OR, odds-ratio; CI, confidence interval; Multiple logistic regression final model was executed on all these statistically significant variables, included together in the model, and selected with the backward stepwise method.

Nsp14 mutations have significant impact on increased genomic mutation density

We then examined the effects of each mutation on genomic mutation density to see whether the relationship between the mutations and MoE status are indicative of a genome-wide trend. Due to selection potentially effecting nonsynonymous mutations differentially, we separated the mutations in the two categories and calculated mutation density separately for each category. Our results show that nsp14 mutations show the most consistent association with mutations between MoE and the whole genome. All three nsp14 mutations (18060C>T, 18736T>C and 18877C>T) which have a significant association with MoE status also show a similar relationship with genomic mutation density (Fig. 2). 18060C>T (L7L) has the lowest odds ratio for MoE status (Table 4), and while it shows a slower increase in synonymous mutation density compared to wildtype isolates (Fig. 2A), it has a significant impact on faster mutation density increase in nonsynonymous mutations (Fig. 2B). In comparison, 18877C>T (L270L) (Figs. 2C and 2D) and 18736T>C (F233L) (Figs. 2E and 2F) both show a high prediction capacity for MoE and an increased mutation density. In comparison, mutations in nsp7 (Figs. S3 and S4) and nsp12 (Figs. S5–S8) show much lower impact on altered mutation density increase rate. 12073C>T, an nsp7 mutation, displays high divergence from wildtype isolate patterns; however, its low sample size (n = 16) creates a skewed distribution of isolates across time, complicating any potential inference.

Figure 2 The distribution of synonymous and nonsynonymous mutations in isolates carrying nsp14 mutations compared to wildtype isolates.

(A and B) Isolates carrying the synonymous 18060C>T mutation (n = 1,585). (C and D) Isolates carrying the synonymous 18877C>T mutation (n = 893). (E and F) Isolates carrying the nonsynonymous 18736T>C mutation (n = 236). Wildtype isolates in all graphs carry the reference nucleotide for the nine positions of interest (11916, 12073, 13536, 13730, 13862, 14408, 18060, 18736, 18877) (n = 5,910). Correlation scores are calculated using Spearman rank correlation.

Conclusions

Our previous work identified RdRp mutations as contributors to the evolution of the SARS-CoV-2 genome and this study confirmed those findings. Furthermore, we hypothesized that mutations of the other critical components of the viral replication and transcription machinery may have similar effects. Our results implicate nsp14 as a source of increased mutation rate in SARS-CoV-2 genomes. Three of the five most common nsp14 mutations, namely 18060C>T, 18736T>C and 18877C>T are associated with increases in both genome-wide mutational load, as well as MoE status, an alternative indicator of mutational rate and virus evolution. Interestingly all three are located within the ExoN domain, which is responsible for the proofreading activity of nsp14; however, only 18736T>C mutation is non-synonymous (F233L), while 18060C>T and 18877C>T are synonymous mutations and therefore, only after functional studies it will be possible to understand their effects on viral replication processes.

The origins and fates of the three nsp14 mutations are also quite different: Being present in the first case detected in the Washington state of the US in mid-January, 18060C>T mutation has been almost completely confined to the US, as 1,657 of 2,007 isolates (82.6%) originate from the US (https://bigd.big.ac.cn/ncov/variation/annotation/variant/18060, accessed 6 September 2020). On the other hand, 18877C>T mutation arising around at the end of January likely in Saudi Arabia and being detected in much less cases (n = 893), is still present in many isolates, most frequently in Saudi Arabia (54.1%) and Turkey (37.4%). 18736T>C mutation was first detected in the US at the beginning of March and like the 18060C>T mutation, has almost completely been limited to the US (281/362 or 77.6%). Unlike the other two, this mutation has been detected in only two isolates since 27 May, and not after 1 July 2020. However, it should be noted that 18877C>T mutation arose within the dominant 23403A>G/14408C>T lineage, while the other two nsp14 mutations are in different lineages. Therefore, dominance or disappearance of different nsp14 mutations may have less to do with these particular mutations and more with the co-mutations. Yet, we cannot rule out possible effects of these nsp14 mutations on the fitness of SARS-CoV-2.

Previous studies on alphacoronavirus nsp14 protein had shown that nsp14, via its exonuclease activity, can modulate host-virus interactions, degrading double-stranded RNA produced during genome replication to suppress immune response, thus increasing viral viability (Becares et al., 2016). SARS-CoV-2 nsp14, due to similar exonuclease activity, is therefore a potential modulator of host interactions, independent of its link to increased mutation load. However, the exact effects of the mutations we identified, two of which are synonymous and may only indirectly affect protein structure, have to be studied experimentally to show any possible changes in viral property that they might affect. Of note, a recent study where codon usage of SARS-CoV-2 was analyzed in terms of temporal evolution of the virus genome revealed that nsp14 is one of three genes (together with S and N genes) that display the highest Codon Adaptation Index (CAI) values (Dilucca et al., 2020a). CAI is a measure of optimal codon usage and indicates how well codons adapt to the host. Based on higher CAI values in nsp14, one could speculate that such mutations have been accumulating preferentially to reach the optimal mutation rate that allows the most advantageous mutation-selection equilibrium for SARS-CoV-2. Indeed, our previous results (Eskier et al., 2020b) indicated that the mutation densities of SARS-CoV-2 genomes are closely related to the pandemic stage and population dynamics directly affects the average mutational load of the viral genome. During the rapid growth stages, such as those observed in March in the UK and the US, replication fidelity can be traded off to gain higher replication rates and broader mutational diversity. However, mutations in the replication machinery that result in too high mutation rates would likely be detrimental and eliminated. On the other hand, a small percentage of the resulting mutations could possibly be advantageous, including those that could confer resistance to antiviral drugs. So far, we or others have not been able to detect such mutations advantageous for the virus, however, higher mutation rates make appearance of such a mutation more likely.

We believe that the mutations discussed in this study can be of help to future studies, in both fighting the COVID-19 pandemic, and better understanding of how mutations in coronavirus replication proteins can affect viral viability and replication fidelity in hosts. Also, it is yet to be determined whether COVID-19 cases infected with SARS-CoV-2 that has mutation(s) that are associated with higher mutation rate respond better to nucleoside analogs, such as remdesivir or ribavirin.

Supplemental Information

Supplemental Information 1 Supplemental Figures.

Click here for additional data file.

The authors would like to thank Mr. Alirıza Arıbaş from Izmir Biomedicine and Genome Center for his technical assistance. The authors also would like to extend their thanks to Izmir Biomedicine and Genome Center (IBG) COVID19 platform IBG-COVID19 for their support in implementing the study.

Additional Information and Declarations

Competing Interests

Author Contributions

Data Availability

Aslı Suner and Gökhan Karakülah are Academic Editors at PeerJ.

Doğa Eskier conceived and designed the experiments, performed the experiments, analyzed the data, prepared figures and/or tables, authored or reviewed drafts of the paper, and approved the final draft.

Aslı Suner conceived and designed the experiments, performed the experiments, analyzed the data, prepared figures and/or tables, authored or reviewed drafts of the paper, and approved the final draft.

Yavuz Oktay conceived and designed the experiments, performed the experiments, analyzed the data, prepared figures and/or tables, authored or reviewed drafts of the paper, and approved the final draft.

Gökhan Karakülah conceived and designed the experiments, performed the experiments, analyzed the data, prepared figures and/or tables, authored or reviewed drafts of the paper, and approved the final draft.

The following information was supplied regarding data availability:

The data is available at Mendeley: Eskier, Doğa; Suner, Aslı; Oktay, Yavuz; Karakülah, Gökhan (2020), “SARS-CoV-2 GISAID isolates (2020-06-17) genotyping VCF”, Mendeley Data, v1. DOI 10.17632/63t5c7xb4c.1

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
