# Peer review of "Mutations of SARS-CoV-2 nsp14 exhibit strong association with increased genome-wide mutation load"

_PeerJ, doi:10.7717/peerj.10181_

## Round 0.1 · original submission · Major Revisions

Both reviewers are very positive and their suggestions can substantially improve your manuscript. I hope that you can perform necessary changes very quickly and re-submit your paper soon.

·

Basic reporting

In this bioinformatics work the authors analyze current SARS-COV-2 libraries and identify that the most frequently observed mutations
for four proteins in the RNA synthesis complex over time in comparison to wildtype
isolates. Specifically, their results suggest mutations in nsp14, an error-correcting exonuclease protein, have the strongest association with increased mutation load. The findings are important but one can wonder the following :
1. All exonucleases have a high error rate and vary significantly from organism to organism. Therefore one wonders why to expect different in viruses. Which is the mutation rate per Kbp or Mbp?
2. How these findings compare with current findings for D614G mutation in SARS-CoV-2 which is becoming infamous for its rising dominance worldwide?

Experimental design

Methodologies
The authors in the design and description of their methods need to be more analytical and exact.
I list below some points
'... obtained from the GISAID
77 EpiCoV database (date of accession: 17 June, 2020) were filtered to remove low-coverage or incomplete genomes, aligned against the reference genomic sequence for SARS-CoV-2'. Which is the reference genomic sequence? How alignment was made.
In general in many cases, their methods section is not complete and self-explanatory.

Validity of the findings

The validity of the findings and results will be better established after the above corrections. In addition, it is not really clear this increased mutation load of it offers to the virus evolutionary advantage also? What about interactions with host?
https://www.ncbi.nlm.nih.gov/pmc/articles/PMC4934755/

Additional comments

The authors need to explain better some aspects of their work as described above and provide better solid evidence for the original nature of their findings and comparison with current advances almost every day in the field.
For example, see also for other coronavirus
https://www.pnas.org/content/112/30/9436

·

Basic reporting

The COVID-19 pandemic has mobilized researchers worldwide in a somewhat unprecedented way. This article uses currently available literature to provide information that can help elucidate the continuing evolution of the SARS-CoV-2 virus and assessing viral strains' mutability. The paper's structure adheres to the current journal's standards and uses concise and professional language.

Experimental design

The methodology followed throughout the paper is precise, and the consideration of calculating mutation densities for non-synonymous and synonymous mutations separately is a positive feature. Moreover, the idea of geographical location as a factor to consider while studying the association of each mutation with the presence of MoE mutations is an interesting one, but as stated, briefly in the paper itself, sequencing efforts may influence the results.

Validity of the findings

In summary, though, the information produced by the current article is of high scientific interest, and after some minor revisions, the paper is recommended for publication.

Additional comments

The above mentioned minor revisions include:
Line 40: As of which date are the numbers regarding death and infection relevant? Is it the same as the day the data was obtained from the database?
Lines 132-133: Additional information would be helpful, especially since the N structural protein’s gene is now firstly mentioned in the article. Maybe the authors can specify the differences in the mutational load?
The geographical location data could be addressed more in the conclusions segment, while restating the biases they mentioned in the text before.

---

## Round 0.2 · Minor Revisions

Thank you again for your submission to PeerJ. I think that you will be able to perform these very minor changes quickly.

1) Improve / extend Figure legends

2) Replace the mentioned by the reviewer reference to this one:
Dilucca M, Forcelloni S, Georgakilas AG, Giansanti A, Pavlopoulou A.
Codon Usage and Phenotypic Divergences of SARS-CoV-2 Genes.
Viruses. 2020 Apr 30;12(5):498.

·

Basic reporting

The revised manuscript stands much better and the authors have addressed almost all of my concerns. The interaction with host importance , I think it is underestimated.

Experimental design

The design and description is no better and more concise.

Validity of the findings

The findings are valid.

Additional comments

The revisions have improved the quality and clarity of the manuscript. I am still though sceptical on the importance of their findings regarding the interaction with host.
In addition, the Figure legends are way too small.
Iam not sure also of this reference can be accepted as a regular citing document since it is in the bioRxiv.

Dilucca M, Forcelloni S, Georgakilas AG, Giansanti A, Pavlopoulou A. 2020a. Temporal evolution and adaptation of SARS-COV 2 codon usage. bioRxiv:2020.05.29.123976. DOI: 10.1101/2020.05.29.123976.

·

Basic reporting

The authors of the article "Mutations of SARS-CoV-2 nsp14 exhibit strong association with increased genome-wide mutation load" have made a significant effort to address the comments and revisions I suggested.

Experimental design

Experimental design is scientifically sound

Validity of the findings

Validity of the findings is OK

Additional comments

I think that the article "Mutations of SARS-CoV-2 nsp14 exhibit strong association with increased genome-wide mutation load" has been significantly improved and therefore is ready to be published

---

## Round 0.3 · accepted · Accept

Thank you for your submission.